

# Physiological aspects and energetic contribution in 20s:10s high-intensity interval exercise at different intensities

Gabriel V. Protzen[1], Charles Bartel[1,4], Victor S. Coswig[2], Paulo Gentil[3] and Fabricio B. Del Vecchio[1]

[1] Physical Education College, Federal University of Pelotas, Pelotas, Rio Grande do Sul, Brazil
[2] Faculty of Physical Education, Federal University of Para, Castanhal, Pará, Brazil
[3] Faculty of Physical Education and Dance, Federal University of Goias, Goiânia, Goiás,
[4] Physical Education Center Admiral Adalberto Nunes, Brazilian Navy, Rio de Janeiro, Brazil

Corresponding author
Gabriel V. Protzen,
gprotzen@gmail.com

## ABSTRACT

**Background**. One of the most popular high-intensity interval exercises is the called "Tabata Protocol". However, most investigations have limitations in describing the work intensity, and this fact appears to be due to the protocol unfeasibility. Furthermore, the physiological demands and energetic contribution during this kind of exercise remain unclear.

**Methods**. Eight physically active students ($21.8 \pm 3.7$ years) and eight well-trained cycling athletes ($27.8 \pm 6.4$ years) were enrolled. In the first visit, we collected descriptive data and the peak power output (PPO). On the next three visits, in random order, participants performed interval training with the same time structure (effort:rest 20s:10s) but using different intensities (115%, 130%, and 170% of PPO). We collected the number of sprints, power output, oxygen consumption, blood lactate, and heart rate.

**Results**. The analysis of variance for multivariate test (number of sprints, power output, blood lactate, peak heart rate and percentage of maximal heart rate) showed significant differences between groups ($F = 9.62$; $p = 0.001$) and intensities ($F = 384.05$; $p < 0.001$), with no interactions ($F = 0.94$; $p = 0.57$). All three energetic contributions and intensities were different between protocols. The higher contribution was aerobic, followed by alactic and lactic. The aerobic contribution was higher at 115%PPO, while the alactic system showed higher contribution at 130%PPO. In conclusion, the aerobic system was predominant in the three exercise protocols, and we observed a higher contribution at lower intensities.

# INTRODUCTION

High-intensity interval exercise (HIIE) is repeated efforts with intensity above 90% of the intensity related to maximal oxygen consumption ($iV \cdot O_{2MAX}$) followed by active or passive recovery (*Buchheit & Laursen, 2013b*; *MacInnis & Gibala, 2017*). Effort and recovery duration and intensity are the mainly manipulated variables during HIIE, which distinctly

affect acute and chronic metabolic responses (*MacInnis & Gibala, 2017*). However, the inconsistency and variability in HIIE protocols may limit its external validity and the data extrapolation to different populations (*Viana et al., 2018b*).

One of the most popular HIIE structure (*Tabata, 2019*) was proposed by *Tabata et al., (1996)*, which is also one of the most inconsistently applied protocols (*Gentil et al., 2016*). It is seven to eight repetitions of 20 s of effort and 10 s of passive recovery (20s:10s) performed until the participant was unable to keep at least 85 rpm. This HIIE model is at an intensity equivalent to 1.7 times the measured $V \cdot O_{2MAX}$, which was calculated by the extrapolation of the linear relationship between submaximal exercise intensity and oxygen uptake (7-8x 20s @170$V \cdot O_{2MAX}$: 10s of passive recovery). Previously, in this type of HIIE, acute studies observed high oxygen consumption (*Viana et al., 2018c*), elevated glycolysis, pronounced glycogen depletion (*Scribbans et al., 2014*), and high parasympathetic inhibition (*Schaun & Del Vecchio, 2018*). Intermittent efforts at 170% of $iV \cdot O_{2MAX}$, obtained with a graded exercise test, are indicated for sprint interval training or repeated sprint training, but not for short HIIE (*Buchheit & Laursen, 2013b*). In a study carried out with physically active young men on magnetic bikes, *Viana et al. (2018c)* verified that the 170% of $iV \cdot O_{2MAX}$ intensity allowed an average of only four repetitions, and induced a very short period at high oxygen consumption rates. However, to date, the effect of this type of exercise in highly trained cycling athletes habituated to this exercise model is unknown.

Notwithstanding, HIIE models based on *Tabata et al. (1996)* have been widely used to improve the metabolic profile or increase physical fitness (*Bonafiglia et al., 2017*; *Domaradzki et al., 2020*; *Logan et al., 2016*; *Ma et al., 2013*; *McRae et al., 2012*; *Scribbans et al., 2014*). However, many investigations (*Domaradzki et al., 2020*; *Logan et al., 2016*; *Ma et al., 2013*; *Scribbans et al., 2014*) have limitations in describing effort intensity, using all-out efforts or different from the intensity corresponding to 170% of $iV \cdot O_{2MAX}$, therefore, differs from the original protocol (*Tabata et al., 1996*; *Viana et al., 2018a*).

Regarding energetic responses, the authors claimed that the 20s:10s protocol reached maximal aerobic and anaerobic demands (*Tabata, 2019*; *Tabata et al., 1996*). This statement was because, at the end of the protocol, the subjects reached their maximal accumulated oxygen deficit (MAOD) and an oxygen uptake equal to their $V \cdot O_{2MAX}$. However, the model applied has been previously questioned (*Bangsbo, 1992*), and the acute physiological impact of different intensities in 20s:10s HIIE on cardiorespiratory and neuromuscular variables is unknown. Information about the contribution of energetic systems during the 20s:10s protocol is essential to understand its physiological demands, and to date, we are aware of no studies have analysed such responses.

Therefore, considering that this knowledge is relevant to the exercise organisation, as it allows to drive physiological stimuli according to the training status, the objectives of the present study were to measure the physiological demands (oxygen uptake, heart rate, and blood lactate), to assess the contribution of energy systems, as well as neuromuscular parameters (total sprint number, mean, and maximum power output) in the 20s:10s HIIE protocol in three different intensities, in cycling athletes and non-athletes.

## MATERIALS & METHODS

### Experimental approach to the problem

The study involved four visits, with minimum rest of 48 h and a maximum of 72 h between them. Participants were instructed not to ingest caffeine or alcohol and not practice intense physical exercises in the 48 h before each trial. On the first day, the ethical research aspects, body mass measurement (Soehnle®, Backnang, Denmark), and height (Standard, Sanny®, São Paulo, Brazil) were collected. On the same day, participants completed an incremental maximal effort test to identify: (i) maximal oxygen consumption ($V \cdot O_{2MAX}$); (ii) maximum heart rate ($HR_{MAX}$); (iii) power associated with the maximum oxygen consumption ($pV \cdot O_{2MAX}$) for sample characterisation and determination of training load.

In the following days, the participants performed the HIIE training sessions in three different intensities in a random order, separated by a minimum and maximum interval of 48 to 72 h, respectively. Subjects performed the training in a mechanical braking cycle ergometer (Biotec 2100, Cefise®, São Paulo, Brazil). Specific software was used to calculate the power output based on wheel speed and previously informed load (Ergometric 6.0, Cefise®, São Paulo, Brazil). The sessions were conducted by previously trained researchers, following the institutional safety manual.

### Subjects

We based the sample size calculation on data from *Lopes-Silva et al. (2015)*. The authors observed that the mean difference for aerobic contribution during HIIE between the two conditions (placebo or substance) was 3%, with standard deviation from means of 2.5%. Seven individuals per group would be required, when assuming 80% power and 5% significance level in a two-tailed test. Considering sample loss of 10%, eight individuals participated per group. The inclusion criteria were: (i) to be physically active (more than 150 min of physical activity per week); (ii) have no musculoskeletal, respiratory, or cardiovascular problems; (iii) non-smoker; (iv) declare not to be anabolic androgenic steroids user. Besides, cyclists should: (i) practice road cycling or mountain bike marathon for more than two years and (ii) have a relative $V \cdot O_{2MAX}$ equal to or greater than 50 mL. $kg^{-1} \cdot min^{-1}$. The sample consisted of 16 males, of whom eight sports-engaged physical education students (age: $21.8 \pm 3.7$; body mass: $65.5 \pm 5.4$ kg; $HR_{MAX}$: $192.7 \pm 2.6$ bpm; $V \cdot O_{2MAX}$: $3375.5 \pm 331.4$ mLO$_2$ $\cdot min^{-1}$; $pV \cdot O_{2MAX}$: $238.1 \pm 18.0$ W) and eight road cycling or mountain-bike athletes (age: $27.8 \pm 6.4$; body mass: $70.3 \pm 9.5$ kg; $HR_{MAX}$: $188.4 \pm 4.5$ bpm; $V \cdot O_{2MAX}$: $4122.2 \pm 506.7$ mLO$_2$ $\cdot min^{-1}$; $pV \cdot O_{2MAX}$: $351.8 \pm 42.6$ W). All cyclists reported doing at least eight hours of weekly training. All subjects filled informed consent, and the Federal University of Pelotas Research Ethical Committee approved the research project (protocol number 77729517.1.0000.5313).

### *Incremental maximal effort test*

The incremental test started with a 2-min continuous warm-up with 0.5 kgf load. After warming up, we increased the load every minute by 0.25 kgf, which corresponds to approximately 25 w, until the participant reached exhaustion, or was not able to maintain the minimum cadence of 90 rpm.

### Training protocols

The experimental sessions began with a 5-min warm-up using a 0.5 kgf load with a cadence between 90 and 100 rpm.

The training protocol followed the previously published pattern of effort:pause of the 20s:10s proposed by *Tabata et al. (1996)*, and was performed using the following intensities: (i) 115%; (ii) 130%; (iii) 170% of iV·$O_{2MAX}$(denominated 115%PPO, 130%PPO, and 170%PPO, respectively), with a cadence control between 90 and 100 rpm. The subjects were oriented to perform as many sprints as possible and to remain seated on the bicycle. The handlebar and saddle were individually adjusted, and we used the same settings in all experimental sessions. The training was interrupted when the subject could not maintain the minimum predefined cadence of 90 rpm or declared voluntary exhaustion.

### Gas exchanges collection

The V·$O_2$ was estimated at every three breaths using an open-circuit gas analyser (V·O2000$^{TM}$, Medical Graphics, Minnesota, US), previously calibrated and following the manufacturer's guidelines. In order to collect gas exchange, a Neoprene$^{TM}$ mask with a high flow pneumotachograph was connected by an umbilical to V·O2000.

Gas exchanges were recorded with the participant sitting on the cycle ergometer for 5 min (*Campos et al., 2012*), to measure oxygen consumption associated with relative rest before the maximal incremental test. This monitoring also occurred during the test to identify the V·$O_{2MAX}$, which was considered as the mean of last-minute values. During training, were collected V·$O_2$ data in three different moments in each exercise protocols: (i) relative rest, (ii) during the whole activity, and (iii) after the end of the exercise, for seven minutes (*Brooks & Mercier, 1994*).

### Blood lactate analysis

A sample of 15μL of blood was obtained from the finger, drained to a heparinised capillary, and transferred to a microtube (Eppendorf$^{TM}$) with 30μL of EDTA anticoagulant to measure the blood lactate concentration ([La$^-$]). We performed the analysis on a lactate analyser (YSI 2300$^{TM}$ Stat Plus, Yellow Springs, Ohio, US). Blood was collected in relative rest and minutes 1, 3, 5, and 7 after the end of each training protocols, in order to obtain the peak lactate concentration ([La$^-$]$_{PEAK}$).

### Heart rate data collection

The heart rate was measured by a specific monitor (V800$^{TM}$, Polar Electro, Kempele, FI), previously validated (*Giles, Draper & Neil, 2016*). Participants wore a thoracic strip with a cardiac sensor and data recorded in the watch. These procedures were used during the incremental test to obtain maximal heart rate (HR$_{MAX}$) and during each trial (115%PPO, 130%PPO, and 170%PPO) to obtain the peak heart rate (HR$_{PEAK}$) in order to characterise the cardiovascular requirements of an exercise model.

### Calculation of energy system contributions

We presented the energy contribution in relative (%) and absolute (kilocalories and kilojoules) values. We assumed that one litre of oxygen is calorically equivalent to 20.9

kilojoules and 5 kilocalories. Aerobic energy was estimated using the $V \cdot O_2$ during exercise protocols subtracted by the $V \cdot O_{2rest}$ through the trapezoidal method (*Bertuzzi et al., 2007*). The $V \cdot O_{2rest}$ was obtained five minutes before the beginning of the protocol, with the subjects seated. The difference between the $[La^-]_{peak}$ and the $[La^-]_{rest}$ was used in the equation to estimate the energy production from the anaerobic lactic system, assuming that the accumulation of 1 mmol.L$^{-1}$ is equivalent to 3 ml.$O_2$.kg$^{-1}$ of body mass (*Di Prampero & Ferretti, 1999*; *Margaria et al., 1963*). The fast component of the excess post-exercise oxygen consumption (EPOC$_{fast}$), which is similar to maximal accumulated oxygen deficit (*Zagatto et al., 2019*), was used to estimate the production of alactic energy (*Di Prampero & Ferretti, 1999*; *Haseler, Hogan & Richardson, 1999*). We observed that the slow component of the bi-exponential model was insignificant. Therefore, we used the monoexponential model (Eq. (1)), and the estimation was calculated by the integration of the exponential part (Eq. (2)). These procedures were applied in previous research (*Bertuzzi et al., 2007*; *Campos et al., 2012*) and follow previously described assumptions (*Di Prampero & Ferretti, 1999*). We used specific software (GEDAE-LaB, São Paulo, Brazil) to calculate the energetic contributions. The procedure was tested and validated against traditional calculations, with an intraclass correlation coefficient of 0.94 for energy expenditure and energy contribution calculation (*Bertuzzi et al., 2016*). Following, we provided the equations used by the software and their respective description; according to *Bertuzzi et al. (2016)*.

$$V \cdot O_{2(t)} = V \cdot O_{2rest} + A[e^{-(t/t)}] \tag{1}$$

$$AL_{ENERGY} = A.\tau \tag{2}$$

where $AL_{ENERGY}$ is alactic system contribution estimated by the fast component of excess post-exercise oxygen consumption, $V \cdot O_{2(t)}$ is the oxygen uptake at time t, $V \cdot O_{2rest}$ is the rest oxygen uptake, $A$ is the amplitude, and $t$ is the time constant.

### Statistical analysis

A Shapiro–Wilk test confirmed the data normality distribution, and we present data as mean and standard deviation (SD). Independent t-tests compared subjects' physical characteristics. We compared training protocols, energy systems, and power output with a two-way analysis of variance (variable × group) with repeated measures. We tested the sphericity of the data by Mauchly's test, and, when violated, applied Greenhouse-Geisser correction. Bonferroni *posthoc* identified the significant differences in power output, and Scheffé *posthoc* determined the differences between training protocols and energy systems.

## RESULTS

Independent t-tests indicated that athletes had lower maximal heart rate ($p = 0.04$) and higher absolute oxygen uptake ($p = 0.02$), age ($p = 0.04$), and peak power output ($p < 0.001$) than non-athletes.

The analysis of variance for the multivariate test (number of sprints, power output, blood lactate, peak heart rate, and percentage of maximal heart rate) resulted in significant differences between groups ($F = 9.62$; $p = 0.001$) and intensities ($F = 384.05$; $p < 0.001$),

**Table 1   Descriptive data from mechanical and physiological variables ($n = 16$).**

| Variable | Non-athletes ($n = 8$) | | Athletes ($n = 8$) | | Total | | Group | Intensity | Interaction |
|---|---|---|---|---|---|---|---|---|---|
| | Mean | ±SD | Mean | ±SD | Mean | ±SD | F (p) | F (p) | F (p) |
| Number of sprints (reps)[*] | | | | | | | 1.2 (0.28) | 173.3 (<0.001) | 0.2 (0.79) |
| 115%PPO | 17.13 | ±3.60 | 15.50 | ±3.34 | 16.31 | ±3.46 | | | |
| 130%PPO | 9.13 | ±2.53 | 8.13 | ±1.81 | 8.63 | ±2.19 | | | |
| 170%PPO | 4.63 | ±1.60 | 3.88 | ±1.25 | 4.25 | ±1.44 | | | |
| Mean power output (w)[*] | | | | | | | 18.2 (0.001) | 104.5 (<0.001) | 4.0 (0.65) |
| 115%PPO | 339.75 | ±28.69 | 411.25 | ±52.21 | 375.50 | ±54.95 | | | |
| 130%PPO | 392.88 | ±44.05 | 515.38 | ±60.18 | 454.13 | ±81.22 | | | |
| 170%PPO | 503.00 | ±50.50 | 653.38 | ±110.10 | 578.19 | ±113.48 | | | |
| Blood lactate (mmol.L$^{-1}$) | | | | | | | 13.9 (0.002) | 4.46 (0.021) | 1.9 (0.16) |
| 115%PPO | 12.07 | ±0.82 | 14.37 | ±1.53 | 13.22 | ±1.68 | | | |
| 130%PPO | 12.99 | ±1.53 | 13.66 | ±0.88 | 13.32 | ±1.25 | | | |
| 170%PPO | 11.28 | ±1.60 | 13.07 | ±1.21 | 12.17 | ±1.65 | | | |
| Peak heart rate (bpm)[#] | | | | | | | 2.3 (0.15) | 309.6 (<0.001) | 0.3 (0.74) |
| 115%PPO | 180.88 | ±9.19 | 187.50 | ±8.72 | 184.19 | ±9.30 | | | |
| 130%PPO | 180.63 | ±9.78 | 185.25 | ±9.15 | 182.94 | ±9.45 | | | |
| 170%PPO | 174.13 | ±12.32 | 178.00 | ±10.45 | 176.06 | ±11.22 | | | |
| Heart rate (%HRmax) | | | | | | | 1.5 (0.24) | 162.2 (<0.001) | 0.02 (0.98) |
| 115%PPO | 93.88 | ±4.63 | 99.49 | ±4.02 | 96.68 | ±5.09 | | | |
| 130%PPO | 93.76 | ±5.28 | 98.28 | ±4.17 | 96.02 | ±5.16 | | | |
| 170%PPO | 90.37 | ±6.27 | 94.41 | ±4.34 | 92.39 | ±5.61 | | | |

**Notes.**

[*]all intensities are different between them ($p < 0.001$)

[#]difference between 115% and 170% ($p < 0.001$)

with no interactions. The results of univariate tests indicated that the number of sprints and peak heart rate were higher in lower intensities, while peak power output was higher at higher intensities (Table 1). The mean duration of each protocol was: 488, 258 and 127 s, for 115%PPO, 130%PPO, and 170%PPO, respectively. Besides, athletes reached higher levels of blood lactate concentration than non-athletes.

Considering absolute contribution (Fig. 1C), there were significant differences between energetic systems ($F = 20.86$; $p < 0.001$) and intensities ($F = 12.65$; $p = 0.001$), with no interactions between systems and groups, intensities and groups, nor systems × intensities × groups. Pairwise comparisons using Bonferroni *posthoc* test localised differences between the three energetic systems, with a $p$-value lower than 0.01, as well as between the three intensities, with $p$-values lower than 0.001. Repeated measures showed a linear trend for decrease the participation of energetic system contribution (aerobic, lactic, and alactic) for kcal ($F = 98.49$; $p < 0.001$), kJ ($F = 98.48$; $p < 0.001$) and litres of $O_2$ ($F = 98.45$; $p < 0.001$), as well as a linear reduction of kcal amount ($F = 110.2$; $p < 0.001$), kJ amount ($F = 110.22$; $p < 0.001$) and litres of $O_2$ consumed ($F = 109.87$; $p < 0.001$) considering all intensities (115%, 130%, and 170%). For the total amount of kcal, kJ, and litres of $O_2$, no differences were found between non-athletes and athletes, but there were significant

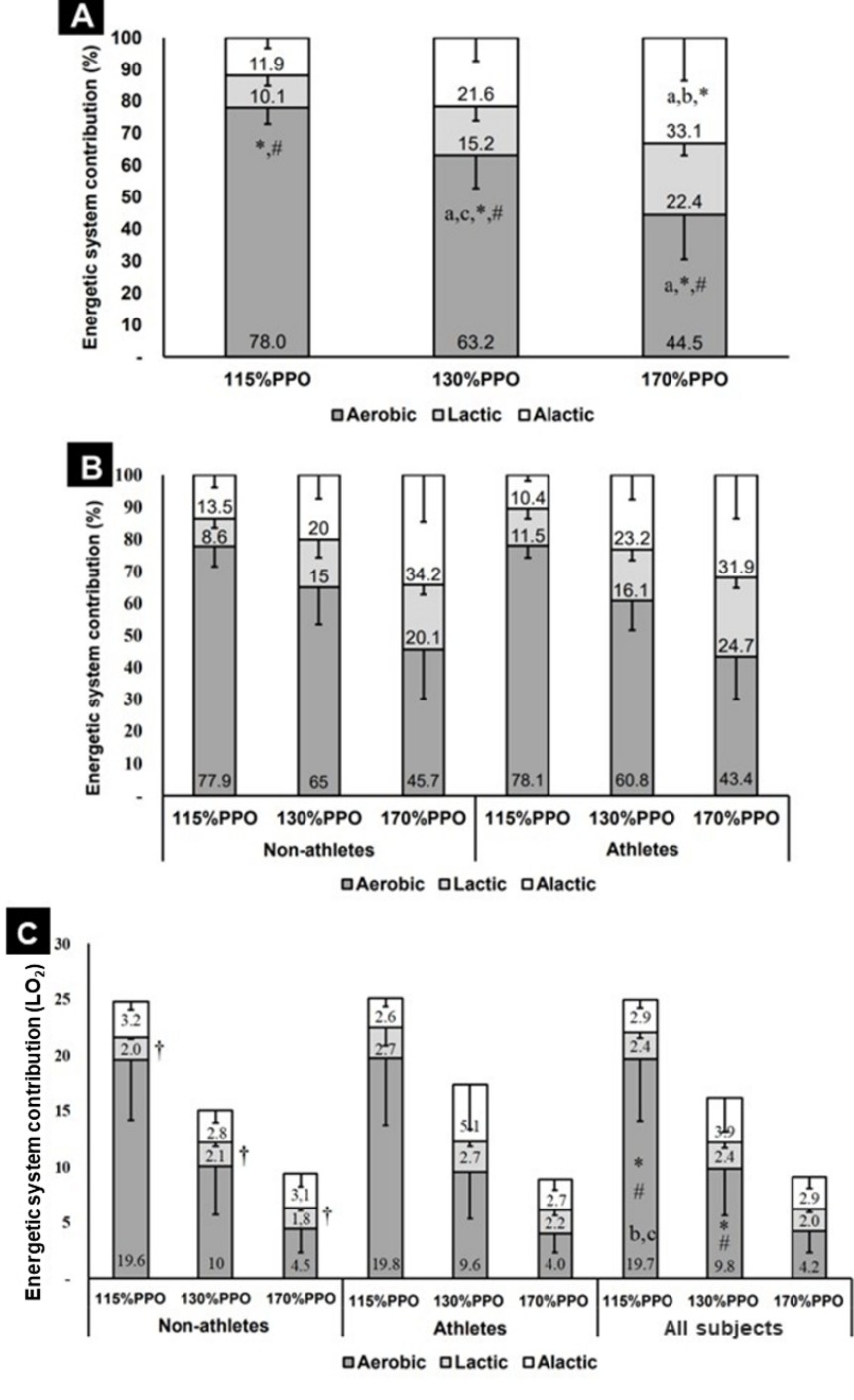

**Figure 1 Energetic system contribution during the 20s:10s protocol at different intensitities ($n = 16$).**
(A) The relative contribution of the energetic system in all subjects. (B) Comparison between athletes and non-athletes of the relative contribution of the energetic system. (C) The absolute contribution of the energetic systems in non-athletes, athletes, and all subjects. * = different from lactic contribution; # = different from alactic contribution; a, b, c = different from 115%PPO, 130%PPO, and 170%PPO respectively, considering each energetic system; † = different from cycling athletes, for the same intensity and energetic system; PPO = peak power output.

differences between intensities ($F = 22.81$; $p < 0.001$), with no interactions between groups and intensity. As a partial analysis by each energetic system, in total amount were found a linear trend for intensity in kcal ($F = 106.52$; $p < 0.001$), kJ ($F = 106.52$; $p < 0.001$) and litres of $O_2$ ($F = 109.9$; $p < 0.001$), with $p$-values lower than 0.001 for all comparisons (115%PPO vs 130%PPO; 115%PPO vs 170%PPO and 130%PPO vs 170%PPO).

Concerning relative energetic contribution system, there was no effect of group (non-athletes vs cycling athletes), but there were significant differences between intensities ($F = 39.3$; $p < 0.001$), and systems ($F = 411.0$; $p < 0.001$), with no significant interactions between group and intensity, group and energetic system. There was no interaction between intensity and energetic systems contribution ($F = 47.81$; $p < 0.001$). Finally, we found no interaction between group, intensity, and energetic system.

Considering the intensity and energetic systems (Fig. 1A), at 115%PPO and 130%PPO, the aerobic contribution was higher than lactic ($p < 0.001$) and alactic ($p < 0.001$), with no difference between lactic and alactic. At 170%PPO, the aerobic contribution was different from lactic ($p < 0.001$) and alactic ($p = 0.02$), and lactic was different from alactic ($p = 0.04$). Additionally, the aerobic contribution was different considering the three selected intensities, 115%PPO was higher than both 130%PPO ($p < 0.001$) and 170%PPO ($p < 0.001$), and 130%PPO was higher than 170%PPO ($p < 0.001$). The lactic contribution was higher at 170%PPO in comparison to 115%PPO, and alactic contribution was higher at 170%PPO in comparison to 115%PPO ($p < 0.001$) and 130%PPO ($p = 0.02$).

In absolute values (Fig. 2A), we found significant differences for intensity ($F = 8.35$; $p = 0.05$), with increased $O_2$ consumption at 130%PPO ($p$-value $= 0.005$ in comparison to 115%PPO and 170%PPO), and higher $O_2$ consumption in athletes ($F = 6.20$; $p = 0.02$), with no interactions between intensities and groups ($F = 4.36$; $p = 0.05$). Figure 2B shows that the three intensities reached different relative values to $V \cdot O_{2MAX}$ ($F = 3.25$; $p = 0.05$), and the *post-hoc* test pointed difference between 115%PPO and 130%PPO ($p = 0.008$), with no differences between non-athletes and athletes, nor interactions. Additionally, there was a quadratic trend in these relative values ($F = 11.30$; $p = 0.005$).

## DISCUSSION

The present study aimed to analyse and compare the energetic system contribution in the 20s:10s HIIE with three different intensities in cycling athletes and non-athletes. As main findings, we found: (i) the relative dominance of the aerobic system in all three intensities when compared with lactic and alactic systems; (ii) the inability to perform the initially proposed number of sprints associated with 170%PPO when the load is calculated from graded exercise test; (iii) the 130%PPO promoted higher oxygen consumption, but only in cyclists.

Previously, *Gaitanos et al. (1993)* showed that the lactic system participation reduced significantly from the first to the tenth sprint of 6s:30s, and, at the end of the exercise, the energy was predominantly from the alactic system, followed by an increase in aerobic contribution. *Trump et al. (1996)* applied intermittent exercise with different effort:pause structure (30s:240s). However, the results have a similar reduction in anaerobic systems

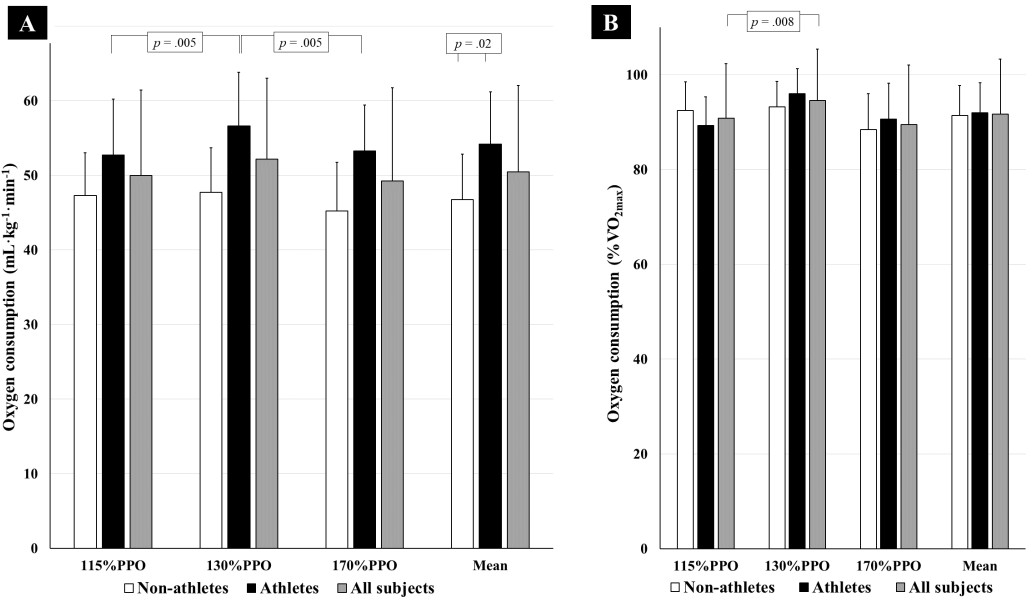

**Figure 2** **Oxygen consumption during the 20s:10s exercise at different intensities ($n = 16$).** (A) Relative to body mass and (B) relative to the maximal oxygen consumption, at 115%PPO, 130%PPO, and 170%PPO, in non-athletes and athletes. PPO = Peak power output.

participation and an increase in the aerobic system participation. Recently, *Panissa et al. (2018)* studied a short-form of HIIE, and their findings also reinforced the raising of aerobic contribution during HIIE. Most of the findings regarding energy systems contributions could be related to training volume, total effort duration, or, specifically in HIIE, the sum of repeated efforts (*Buchheit & Laursen, 2013b*). As these variables increase, the contribution of the aerobic system also rises (*Gastin, 2001*; *Glaister, 2005*; *Trump et al., 1996*).

Based on this, it was not surprising that the protocols showed high aerobic contribution (*Gaitanos et al., 1993*). Even the low exercise duration, found in the 170%PPO condition, had an average length of approximately 120 s. About 1/3 of this time involved pause periods, in which aerobic contribution is very high (*Brooks & Mercier, 1994*). The more substantial aerobic contribution found in the 115%PPO would be explained by the relatively lower required anaerobic contribution for lower exercise intensities. Furthermore, 115%PPO allowed more repetitions, contributing to a longer duration in the session (488 s, vs 258 and 128 in 130%PPO and 170%PPO, respectively). We also observed that, as the intensity increases, the anaerobic systems exert a higher relative, but not absolute, contribution. This higher relative contribution might have occurred because, when exercising at higher intensities, it allowed a lower number of bouts, reducing the total duration of the effort.

Despite the high blood [La-], frequently higher than 11 mmol.L$^{-1}$, the relative glycolytic contribution was not so high, probably since each lactate unit corresponds to only 3 ml.O$_2$.kg$^{-1}$ (*Bertuzzi et al., 2016*; *Bertuzzi et al., 2007*; *Campos et al., 2012*), and the participants had higher oxygen consumption than accumulated lactate, even after respective conversion. This fact reinforces the hypothesis that measuring the glycolytic contribution of a given

exercise considering only the blood [La-] might not be recommended (*Bertuzzi et al., 2016*), or the glycolytic contribution is quite small during some HIIE (*Gaitanos et al., 1993*; *Trump et al., 1996*). Another possibility is that a greater amount of lactate was metabolised within the muscle when the exercise was longer (*Brooks, 1986*); possibly reducing the lactic contribution. It is important to highlight that the assumption of equating 1 mmol.L$^{-1}$ of lactate to 3 ml.O$_2$. kg$^{-1}$ stands for submaximal exercises. However, many other studies used that to assess supramaximal exercises lactic contribution (*Bertuzzi et al., 2007*; *Campos et al., 2012*; *Lopes-Silva et al., 2015*). Further comparisons using direct measurements (i.e., muscle biopsy) should be made to confirm the reliability and replicability of this method.

The athletes presented higher blood [La$^-$] for all intensities, therefore, also presented a higher absolute lactic contribution, this occurred probably due to the greater anaerobic capacity of this population (*Ponorac et al., 2007*), which contributed to the higher power output during the efforts. Regarding oxygen consumption in athletes, the highest value was in 130%PPO. Considering that maintaining higher V·O$_2$ values during exercise is essential for V·O$_{2MAX}$ increase (*Midgley & Mc Naughton, 2006*), the optimal intensity for aerobic power development in athletes, in this time structure, would be near to 130%PPO. Previously, on a treadmill, other studies demonstrated that for 30s:15s training in active young men and 15s:15s in middle-aged runners, the intensities of 110%vV·O$_{2MAX}$ and100%vV·O$_{2MAX}$, respectively, presented the higher oxygen consumption (*Aguiar et al., 2013*; *Billat et al., 2001*). Otherwise, subjects maintained the V·O$_2$ closed to 90–95% of the V·O$_{2MAX}$, which is lower than the 100% reached in the original study and questioned the author's statement that the 'Tabata training' should be considered as ''one of the most energetically effective exercise training protocols for maximally improving both the aerobic and anaerobic energy-supplying systems'' (*Tabata, 2019*).

Interestingly, at the initially suggested intensity of 170% V·O$_{2MAX}$ (*Tabata et al., 1996*), most of the athletes and non-athletes were unable to complete the 7-8 sprints. This aspect was previously questioned (*Viana et al., 2018c*), and these incompatibilities are possibly due to the differences between the tests used to obtain the training loads (*Tabata, 2019*), which put some light in this debatable point from the findings of the present investigation. While *Tabata et al. (1996)* used an obsolete, unpractical and questionable protocol that requires a large number of 10-min visits to the laboratory (*Bangsbo, 1992*; *Medbo et al., 1988*), *Viana et al. (2018c)* we used the traditional, worldwide-used, and very practical maximal graded exercise test (*Buchheit & Laursen, 2013b*).

Even using a similar graded testing protocol, our findings are different from those found by *Viana et al. (2018c)*, in which the intensity of 115%PPO allowed the accomplishment of 7 ± 1 sprints, our results suggest that it is possible to perform this same number of bouts at a higher intensity (130% PPO). Probably this difference is due to the type of ergometer used since previous authors used a magnetic locking model. In contrast, the present study used a mechanical braking device, similar to that used by *Tabata et al. (1996)*. Such differences further reinforce the problem of generalisation of interval protocols reported in previous studies (*Viana et al., 2018c*), especially regarding the protocol proposed by Tabata et al. (*Tabata et al., 1996*; *Viana et al., 2018a*).

There seems to be an apparent conflict in the structural variables of the HIIE protocol with the 20s:10s effort:pause structure as commonly used (*Tabata et al., 1996*). According to previous studies, we classify as a short HIIE model (effort and pause block lasting less than one minute). However, the intensity close to 170%V·O$_{2MAX}$ is characteristic of Sprint Interval Training models (*Buchheit & Laursen, 2013a*). Notwithstanding, the effort:pause ratio for sprint interval training is 1:4-8 due to the need to recover anaerobic pathways to maintain an elevated power output (*Brooks & Mercier, 1994*; *Glaister, 2005*), whereas in the classic 20s:10s protocol this ratio is 2:1. This conflict might explain why it was not possible to reach 7–8 bouts when using 20s:10s, with a load equivalent to 170%V·O$_{2MAX}$, even in athletes.

Finally, despite the methodological issues that we should consider regarding this specific HIIE protocol (*Gentil et al., 2016*; *Tabata et al., 1996*; *Viana et al., 2018c*), its potential to induce positive physiological changes should be recognised (*Ma et al., 2013*; *Miyamoto-Mikami et al., 2018*; *Scribbans et al., 2014*; *Tabata et al., 1996*). Both "classical" studies by the "Tabata" group showed that participants could achieve relevant aerobic and anaerobic improvements in a very high time-efficient manner (*Tabata et al., 1996*). More recently, the same group showed that this protocol could significantly increase aerobic power, maximal accumulated oxygen deficit, and thigh muscle cross-sectional area (*Miyamoto-Mikami et al., 2018*). Further longitudinal studies should investigate this 20s:10s protocol at an intensity range of 115 to 130%PPO in order to raise the time near and at V·O$_{2MAX}$.

## CONCLUSIONS

In conclusion, to a 20s:10s HIIE protocol, the aerobic contribution is predominant, independently of the intensity applied in a range from 115%PPO to 170%PPO. Despite that, the lower the intensity, the higher is the aerobic contribution, and 130%PPO is the suggested intensity to induce higher V·O2 in trained cyclists. Finally, to reach about eight sprints, we propose the intensity of 130% the power at V·O$_{2MAX}$ obtained in a graded test, either for athletes or non-athletes.

## ACKNOWLEDGEMENTS

The authors would like to acknowledge Dr Marlos R. Domingues and Dr Airton Rombaldi for the contributions to the paper and the participants of the study for all commitment.

### Funding

The authors received no funding for this work.

### Competing Interests

The authors declare there are no competing interests.

## Author Contributions

- Gabriel V. Protzen conceived and designed the experiments, performed the experiments, analyzed the data, prepared figures and/or tables, authored or reviewed drafts of the paper, and approved the final draft.
- Charles Bartel, Victor S. Coswig and Paulo Gentil conceived and designed the experiments, authored or reviewed drafts of the paper, and approved the final draft.
- Fabricio B. Del Vecchio conceived and designed the experiments, analyzed the data, prepared figures and/or tables, authored or reviewed drafts of the paper, and approved the final draft.

## Human Ethics

The following information was supplied relating to ethical approvals (i.e., approving body and any reference numbers):

The Federal University of Pelotas Ethical Committee approved this research project (77729517.1.0000.5313).

## Data Availability

All data are available in the Supplementary Files.

## Supplemental Information

Supplemental information for this article can be found online at http://dx.doi.org/10.7717/peerj.9791#supplemental-information.

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
