# Peer review of "Physiological aspects and energetic contribution in 20s:10s high-intensity interval exercise at different intensities"

_PeerJ, doi:10.7717/peerj.9791_

## Round 0.1 · original submission · Major Revisions

I invite the authors to address all the issues raised, and in particular the methodology, as well as to improve the language editing and grammar.

Reviewer 1 ·

Basic reporting

Clear language and adequate structure.

Experimental design

The experimental design is presented clearly.

Validity of the findings

The assumptions and limitations of the methods used should be discussed.

Additional comments

The idea is interesting and useful for practitioners. The authors should consider some serious methodological issues. 1. The method used to estimate energy contribution is based on several assumptions that have not reported in the methods, 2. The duration of each protocol and between groups is not reported and this may affect the recovery curve from which you have calculated the energy contribution. 3. Using this specific ergometer you need to report how power was measured when cadence was not constant (90-100 rpm). Moreover, the results section is full of meaningless numbers that distract reading. I suggest you to insert the numbers in the existing tables and report the actual findings in this section indicating significance or not (ony).

Specific comments
Avoid using the term “Tabata protocol”
Line 41. Explain the abbreviations
Lines 58-59. Is this ergometer valid? How the power was measured?
Line 72-75. Use “:” instead of “=”
Line 83. kgf.m is not the correct units of measure.
Line 83-84. Given the non steady cadence you need to explain how power output was measured with this ergometer
Line 94. Change to “every three breaths”
Lines 119-133. You need to provide equations and assumptions for all calculations. The references you provide are secondary publications. You need to refer the initial works of the approach you have used providing appropriate equations. Moreover, why not using the MAOD instead of EPOC? You need to explain your selection and the limitations of the method in the discussion section. The assumption of equating lactate with oxygen uptake is true during submaximal steady state exercise but not during sprinting.
Line 120. How you have done these calculations. You need to explain.
Line 123. What do you mean? “…expressed as rest values,…”
Line 126. Campos et al. is used this assumption. They have not introduced this assumption which stands only for submaximal exercise.
Line 224. You should report the duration of all testing sessions for each group for a better evaluation of results.
Lines 232-234. Or because a greater amount of lactate was metabolized within the muscle during the longer duration efforts. Discuss it.
Line 266. Do not start a sentence using “In which”

Reviewer 2 ·

Basic reporting

1. Overall there are several grammatical errors throughout the manuscript. The authors are encouraged to perform a thorough edit upon resubmission.

2. The authors have cited previous studies relevant to the current manuscript. However, some recent articles have not been cited:
a. Lines 25-26: Recent cycle training studies using Tabata: https://www.ncbi.nlm.nih.gov/pubmed/28177701; https://www.ncbi.nlm.nih.gov/pubmed/27936084; https://www.ncbi.nlm.nih.gov/pubmed/29733694; https://www.ncbi.nlm.nih.gov/pubmed/26664271

Experimental design

1. Lines 62-65: I commend the authors for performing an a priori sample size calculation, however, could the authors specify the parameters used in the calculation? For instance, what is “aerobic contribution” (oxygen consumption during Tabata protocols?) and what are “the two conditions”. Provide details here so the reader doesn’t have to refer back to Lopes-Silva 2015 to find this information.

2. Lines: 69-70: Was any information collected to confirm that cyclists were actively/currently training? If so please include this as eligibility criteria.

3. Lines 143-144: Can the authors comment on how distinct they think the two groups are? As prefaced in the introduction, the study was designed to compare “non-athletes” and highly trained cyclists. However, mean VO2max was not higher in highly trained athletes and the mean and SD VO2max values (lines 73-75) suggest that some athletes had a lower VO2max than some “non-athletes”.

4. Lines 145-147: The authors should report the specific results from the 2-way ANOVA (i.e. which outcomes had significant effects). Currently, it reads as if all these variables had significant main group and intensity effects, which (according to Table 1) is not true.

Validity of the findings

1. Table 2 is missing a footnote to define the * symbols.

2. Lines: 207-208: I believe main finding #1 pertains to the observation that aerobic contribution was the greatest source of relative energy contribution (Figure 1A), but this main finding is unclear as currently written. Similarly, the authors should clarify main finding number 3. In cyclists, 130%PPO promoted higher VO2 compared to 115% and 170%. As currently written, main finding #3 may be misinterpreted to mean a difference in VO2 between cyclists and non-athletes during the 130% PPO session.

3. Line 228-229: Given the observation that increasing intensity increased relative but not absolute anaerobic contribution, graphically reporting the absolute energy contribution data may help illustrate this finding (instead of in table format)? Showing absolute energy contribution in a bar graph as a separate panel beside relative contribution (Figure 1A) may be a more clear approach to illustrating this point than having the reader compare between Figure 1A and Table 2.

Additional comments

1. Lines 27-29: Can the authors provide more regarding the limitations of previous work? E.g. expand on what is meant by “limitation in describing effort intensity”.

---

## Round 0.2 · Minor Revisions

Some limitations of the study should be discussed. Differences with Tabata's protocol should be also evidenced.

Reviewer 1 ·

Basic reporting

no comment

Experimental design

no comment

Validity of the findings

Several limitations should be reported in the discussion. The selection of the protocol for intensities 115, 130 and 170% calculation is not the same as that used by Tabata et al., (1996). The oxygen debt instead of accumulated oxygen deficit was measured.

Additional comments

The authors have improved the paper. There are some points for further consideration. Several limitations should be reported in the discussion. The selection of the protocol for intensities 115, 130 and 170% calculation is not the same as that used by Tabata et al., (1996). The oxygen debt instead of accumulated oxygen deficit was measured.
Line 98. Was this protocol similar to the protocol used by Tabata et al., (1996) for the measurement of power corresponding to VO2max? This is critical as your comparison is based on this specific Tabata protocol.
Line 106. You have defined iVO2max in the introduction. Is it the same as the pVO2max? How 115, 130, 170% of iVO2max was calculated? By extrapolation of the power vs. VO2 relationship ? or as a percentage of pVO2max. These are not the same. You need to clarify and discuss this as a limitation for the sustainability of repetitions (3-4 in your study, 7-8 in other studies)
Line 112. The duration of each protocol should be reported in the results section.
Line 131. Specify if the peak value was used for the calculations.

---

## Round 0.3 · accepted · Accept

The authors have satisfactorily addressed the issues raised by the reviewers.